# Formalising Anti-Discrimination Law in Automated Decision Systems

## Abstract

We study the legal challenges in automated decision-making by analysing conventional algorithmic fairness approaches and their alignment with anti-discrimination law in the United Kingdom and other jurisdictions based on English common law. By translating principles of anti-discrimination law into a decision-theoretic framework, we formalise discrimination and propose a new, legally informed approach to developing systems for automated decision-making. Our investigation reveals that while algorithmic fairness approaches have adapted concepts from legal theory, they can conflict with legal standards, highlighting the importance of bridging the gap between automated decisions, fairness, and anti-discrimination doctrine.

## 1 Introduction

Automated decision-making using predictive models is becoming increasingly important in many areas of society, including lending [60, 100, 107], criminal justice [31, 14, 151], hiring [64, 59, 25], and welfare eligibility [41, 56, 113]. Instances of large-scale failures, from disproportionately harming vulnerable people in welfare eligibility assessments [113] to bias in consumer lending [80], highlight the need for lawful implementation. Scrutiny of ML-based decisions is heightened by concerns about replicating human biases and historical inequality [97, 41, 93].

Concerns about algorithmic bias have spurred research into *fair* ML. Early discourse on fairness in ML was relatively narrow due to technical constraints [29, 61]. More recently, researchers have developed formal definitions of fairness in algorithmic decisions and methods to measure fairness in predictive models [49, 31, 27, 147, 87, 85]. Algorithmic fairness definitions generally measure prediction disparities across groups with different legally protected characteristics [90, 136, 87, 17]. This research has resulted in several proposals, including statistical metrics to assess the fairness of individual predictive models [136, 111, 22, 24], fairness for model auditing [70, 103, 63, 89, 98], and fairness constraints on models [31, 148, 50, 145, 12].

These criteria simplify fairness into measurements of disparity that do not inherently map to *unlawful* discrimination. The usefulness of these metrics in practice is limited as incomplete or even irrelevant measures for legal investigations. There have been important efforts to bridge the gap between legal and technical approaches to fair ML [81, 55, 51, 144, 139, 1, 46]. Lawyers have highlighted the challenges of the narrow construction of fairness metrics focusing on disparity in predictions rather than more nuanced definitions of discriminatory conduct and the broader context of the automated decision-making process [55, 51, 144, 1]. We aim to contextualise and formalise legal concepts of algorithmic discrimination beyond the narrow construction of statistical disparity.

The predominance of US analysis of fairness and discrimination in ML, lack of non-US ML datasets [78], and the limited legal scholarship translating these concepts, has inadvertently fostered a series of misconceptions that pervade the field. However, very few papers have engaged with

anti-discrimination laws outside of the United States [143, 139, 1, 140, 67, 76]. We aim to introduce new principles and methods to deal with the issues identified in this literature. By avoiding the nuanced legal realities of other jurisdictions, models designed to comply with US laws may breach UK laws or those in comparable jurisdictions. Our paper addresses this gap by providing a rigorous analysis of UK discrimination law, correcting some mischaracterisations, and establishing a more accurate foundation for developing fair ML in the UK and its related jurisdictions.

## 1.1 Automated Decision-Making

Let $x_i \in \mathbb{R}^p$ be a vector of observed attributes for individual $i$. A decision-maker must choose a decision $a \in \mathcal{A}$, where $\mathcal{A}$ is closed. Further, we assume that the decision-maker wants to decide based on a future outcome $y_i \in \mathcal{Y}$ for individual $i$. Here, we assume $\mathcal{Y} = \mathbb{N}$, which can be relaxed.

Decision-making under uncertainty has long been studied in statistical decision theory [108, 32, 13, 99]. Let $u(y, a)$ be a utility function that summarises the *utility* for the decision-maker. The optimal decision is then

$$a^\star = \arg\max_{a \in \mathcal{A}} \sum_{y \in \mathcal{Y}} u(a, y) p(y|a) . \tag{1}$$

The decision-maker usually neither knows $y_i$ nor $p(y|a)$ at the time of the decision. Hence, the decision must be based solely on $x_i$. In an SML setting, a prediction model $\hat{p}(y|x)$ is trained to compute the predicted probability distribution (pmf) $\hat{\pi}_i = \hat{p}(y|x_i)$ for individual $i$, with the support on $\mathcal{Y}$. Further, let $\hat{y}(\hat{\pi}_i) \in \mathcal{Y}$ be the classification made based on $\hat{\pi}_i$. In simple settings, the decision can be formulated as a decision function $d(\hat{\pi}_i) \in \mathcal{A}$ that is used to choose an appropriate action based on $\hat{\pi}_i$. In the binary $y$ and $a$ case, it reduces to a simple threshold $\tau$, i.e., $d(\hat{\pi}) = I(\hat{\pi} \leq \tau)$, where $I$ is the indicator function and $\hat{\pi}_i = \hat{p}(y = 1 \mid x_i)$. We often train a model $\hat{p}(y|x)$ based on previous data $D = (\mathbf{y}, \mathbf{X})$, drawn from a population $p(y, x)$, where both $x_i$ and $y_i$ are known. Replacing $p(y|x_i)$ with the predictive model $\hat{p}(y|x_i)$ in Eq. 1 gives an optimal decision.

## 1.2 Algorithmic Fairness

To define algorithmic fairness, we separate $x_i$ into protected and legitimate features $x_i = (x_{pi}, x_{li})$; we drop $i$ to simplify notation. Here, $x_p \in \mathcal{C}$ indicates protected attributes, with $\mathcal{C}$ being the set of different groups. Legally protected characteristics commonly identified in datasets include gender, race, and age. Many fairness metrics aim to evaluate the fairness of an SML model for commonly identified protected characteristics in datasets, including gender and race [49, 31, 82, 85].

**Statistical parity**, or demographic parity, is one of the central algorithmic fairness metrics [31, 136, 90, 82]. For statistical parity to hold, it requires that

$$\mathbb{E}_x \left[ \hat{p}(y|x) \mid x_p \right] = \mathbb{E}_x \left[ \hat{p}(y|x) \right] , \tag{2}$$

such that the model predictions, in expectation over $x$, need to be the same for the different groups [31, 136]. Given that the decision function $d(\boldsymbol{\pi})$ is the same for the different groups, statistical parity results in equal decisions for the different groups. However, we discuss later in this paper that, in practice, statistical parity may exacerbate inequality or even result in unlawful discrimination [10, 74, 63].

**Conditional statistical parity** extends statistical parity to account for legitimate features $x_l$. The model predictions should only differ across protected groups *to the extent that the difference is conditional on legitimate factors* [31, 136, 23]. This can be formalised as,

$$\mathbb{E}_x \left[ \hat{p}(y|x) \mid x_l, x_p \right] = \mathbb{E}_x \left[ \hat{p}(y|x) \mid x_l \right] , \tag{3}$$

so that, conditional on legitimate features $x_l$, there should not be any difference in predictions between groups given by the protected attribute. Below, we discuss the legitimacy of variables that correlate to protected attributes [34, 77].

Other similar group comparison metrics have been proposed, such as error parity, balanced classification rate, and equalised odds [49, 31, 136, 90, 82, 30]. Also, more individual approaches to parity have considered whether otherwise identical individuals are treated differently if they have different protected attributes [34, 68]. Finally, ideas from causal inference and counterfactual analysis have also been proposed to measure outcome consistency for individuals across protected groups [75, 69, 106, 149, 26, 142, 92, 6]

## 1.3 Anti-Discrimination Law

The algorithmic fairness literature largely identifies statistical disparities in predicted outcomes for binary marginalised groups. Legally, discrimination is both broader and more detailed. Not all actions perceived as discriminatory are unlawful, and some non-obvious actions may be prohibited. Anti-discrimination law only applies to select duty-bearers in certain conditions [65]. Individuals' friendship choices being based on race are not legally regulated, despite sometimes seeming unfair [36, 65]. It only applies to protected attributes. An algorithm that rejects a loan application because the applicant uses an Android phone rather than an iOS device may seem unfair because it does not reflect the true default risk but is a proxy for the applicant's income [2, 76]. However, in isolation, this would not be unlawful discrimination under UK law because poverty is not a protected attribute [96].

The prohibition on discrimination traces its legal roots to the Universal Declaration of Human Rights, which established equality and freedom from discrimination as fundamental human rights, further advanced in several international treaties [132, 88], and enacted as legislation worldwide spurred by the Civil Rights Movement [86, 65]. The United Kingdom implemented several anti-discrimination laws in the 20th century [118, 119, 116], which were consolidated in the Equality Act 2010 [40].

The Equality Act protects "age; disability; gender reassignment; marriage and civil partnership; pregnancy and maternity; race; religion or belief; sex; sexual orientation" [40, s 4]. Algorithmic fairness literature has often oversimplified these protected characteristics as simply identifying visible traits when each has complex social meanings [57]. One complexity is, for example, the difference between a person with a protected attribute by biological fact or by identifying with a protected group [73]. UK anti-discrimination law distinguishes between *direct* discrimination and *indirect* discrimination. While analogous to the US disparate treatment and disparate impact doctrine, there are important distinctions, meaning they should not be so easily elided [1].

**Direct discrimination** occurs when an individual is treated less favourably than another based on a protected characteristic [40, s 13]. To establish direct discrimination, it is necessary to identify the specific protected characteristic involved, demonstrate the less favourable treatment (by real or hypothetical comparison), and prove that this treatment was caused "but for" the protected attribute. The intention of the decision-maker is not required or necessary [123, 131].

**Indirect discrimination** refers to a policy, criterion, or practice (PCP) that disproportionately disadvantages a group with a particular protected attribute compared to those without [40, s 19]. To prove indirect discrimination, one must identify such a PCP, show that it puts a group defined by its protected attribute at a particular disadvantage compared to those without such attribute, and evaluate whether it is justifiable as a proportionate means of achieving a legitimate aim.

English common law is either in force or is the dominant influence in 80 legal systems that govern approximately 2.8 billion people, not including the US [28]. UK anti-discrimination law is very similar to numerous Commonwealth and common law jurisdictions, including Australia [3], Canada [20], India [47], New Zealand [91], South Africa [110], and the pending bill in Bangladesh [9]. European Union law also has broadly the same principles and discrimination case law evolved in parallel during the UK's membership [45]. It is increasingly important to gain a nuanced understanding of unlawful discrimination in AI systems as new laws aim to prevent future harms [44, 16].

## 1.4 Contributions and Limitations

This paper makes four core contributions at the intersection of automated decision-making, fairness, and anti-discrimination doctrine.

1. We formalise critical aspects of anti-discrimination doctrine into decision-theoretic formalism.
2. We analyse the legal role of the data-generating process (DGP) and develop the DGP as a theoretical framework to formalise the legitimacy of the prediction target $y$ and the features $x$ in supervised models for automated decisions.
3. Further, we consider the legal and practical effects of approximating the DGP in supervised models. We propose *conditional estimation parity* as a new, legally informed target.
4. Finally, we provide recommendations on creating SML models that minimise the risk of *unlawful* discrimination in automated decision-making.

Our paper is formally limited to analysing and providing novel recommendations for the UK. While we discuss related jurisdictions that are functionally similar and based on English common law,

specific legal advice should be followed with respect to different jurisdictions. Accountability varies by jurisdiction and context, which is why our paper underscores the importance of careful, informed classification by experts with appropriate legal advice.

## 2 Automated Decisions and Discrimination

### 2.1 Legitimacy of True Differences

In SML, it is crucial to differentiate unlawful discrimination from mere statistical disparities and concepts of algorithmic fairness. While formal equality may map to statistical parity, anti-discrimination laws in the UK and related jurisdictions aim to achieve substantive equality. Despite the general rule that individuals should not receive less favourable treatment based on their protected attributes, courts acknowledge that treating all groups the same can actually disadvantage a protected group and minimise important structural and true differences [137, 143]. Therefore, substantive equality may sometimes require legitimate differential treatment because of the true differences among individuals [137, 52, 144, 139].

For instance, insurance decisions that might otherwise be construed as discriminatory – specifically concerning gender reassignment, marriage, civil partnership, pregnancy, and sex discrimination – are permissible if they are based on reliable actuarial data and executed reasonably [40, Sch 9. s 20]. Financial services can also "use age as a criterion for pricing risk, as it is a key risk factor associated with for example, medical conditions, ability to drive, likelihood of making an insurance claim and the ability to repay a loan" [117, para. 7.6]. These exemptions highlight legal recognition that certain group distinctions, particularly those involving risk assessment, are relevant and necessary for the equitable operation of such services. Similar statutory exemptions are found in other similar anti-discrimination laws, including the European Union [43, art 2], Australia [8, s 30-47], Canada [20, s 15], New Zealand [91, s 24-60] and South Africa [110, s 14].

### 2.2 True Data Generating Process

Therefore, an important aspect from the legal perspective that is overlooked in the existing literature is the distinction between a "true data-generating" process (DGP) and the estimated model $\hat{p}(y|x)$. To formalise, we assume that there exists a true DGP, $D \sim p(y, x)$, where $D_i = (y_i, x_i)$. Further, we use $p(y|x_i^{\text{true}})$ to denote the true probability (pmf) for individual $i$, given the true features $x_i^{\text{true}}$.

We make multiple observations on the role of the "true" model and its use in connecting predictive modelling and legal reasoning.

First, understanding the limits of predictive models is crucial to explore inherent uncertainties and limitations in predictions. The true model is, in practice, never observed or known. When developing $\hat{p}(y|x)$, the target is often to select the model with the best predictive performance, which is closely connected to the role of the true DGP [15, 134, 135, 133]. For this reason, the "true" model may include features in $x_i^{\text{true}}$ that are not observed in the data, sometimes referred to as an $\mathcal{M}$-open setting when the "true" model is not included in the set of candidate models [15, 135].

Second, we assume that $p(y|x_i)$ is a probability distribution over $\mathcal{Y}$, introducing some level of aleatoric uncertainty in the true underlying process [95, 58, 114]. This means that perfect prediction of $y_i$ may not be possible, even with knowledge of the true DGP. The distinction between aleatoric and epistemic uncertainty is important from a legal perspective. The reason is simple: the uncertainty coming from estimation is the (legal) responsibility of the modeller, while the aleatoric uncertainty can instead be considered a true underlying general risk.

Third, the true DGP connects to judicial legal reasoning. Courts must engage theoretically with legal and normative conceptions of what is justifiable and what constitutes unlawful discrimination. Judges consider legitimacy, proportionality, and necessity when evaluating actions, and hypothetical alternatives, that led to less favourable treatment. Although, courts are not oracles. Discrimination case law may not pinpoint what the perfect decision should have been. However, courts will engage in a similar theoretical process of reasoning about the decision-making process to the true DGP to understand whether the actions were justified or unlawful. We explain legal reasoning within this framework throughout the paper and in a real-world case on unlawful discrimination in algorithmic decision-making (see Appendix A).

## 2.3 Estimation Parity

Legally, distinguishing between a true difference and an estimated one is important. We approximate the true DGP with a model $\hat{p}(y|x)$ based on training data when training an SML model. The approximation introduces estimation error

$$\epsilon_i = \hat{\boldsymbol{\pi}}_i - \boldsymbol{\pi}_i = \hat{p}(y_i|x_i) - p(y_i|x_i^{\text{true}}). \tag{4}$$

Algorithmic fairness literature often assumes the absence of estimation error [see e.g., 49] or assumes that the true causal structure is known [150, 68, 26, 21]. In practice, this is rarely the case. Hence, it is crucial, both practically and legally, to distinguish between the true underlying probabilities $\boldsymbol{\pi}_i$ and the estimated probabilities $\hat{\boldsymbol{\pi}}_i$. While the true underlying probability may sometimes be defensible (Section 2.2), introducing an estimation error that disadvantages individuals based on protected attributes invokes discrimination liability.

As the model will try to approximate the true data-generating process, modellers' expectations are difficult to ascertain. The law is unlikely to set a deterministic standard that any adverse effects of estimation will make a modeller liable. The modeller should try to approximate the true model as much as possible [see 4, 141, 135, 133, for discussions on model misspecification]. However, where an estimation disparity reaches a threshold for discriminatory effects, the legal evaluation would require analysing the steps taken to test and mitigate estimation disparity (even though the intent is immaterial).

The potential bias in training data presents a risk that the estimation model will introduce bias against individuals with protected attributes (Section 2.6). Historical discriminatory lending practices, for example, could be perpetuated through biased training data [18, 104]. Such biased estimations may introduce biased outcomes that are not reflective of true differences, potentially leading to discriminatory outcomes. Therefore, we introduce "Conditional Estimation Parity" to formalise the legal context of estimation.

**Conditional Estimation Parity** is the difference in estimation error between groups with a protected attribute, given legitimate features, i.e.,

$$\mathbb{E}_x[\epsilon \mid x_p, x_l] = \mathbb{E}_x[\epsilon \mid x_l]. \tag{5}$$

Reducing the error in Eq. 4 is expected to diminish the risk of conditional estimation disparity. However, assessing conditional estimation parity is complex due to inherent challenges in evaluating estimation error.

It is crucial to examine both mathematical and legal causal theories of why certain differences are legitimate bases to make classification distinctions [71]. We examine the mathematical basis for identifying statistical disparities in the context of unlawful discrimination. In Section 2.5, 2.7, and 2.6 we consider the causal relationships between legitimate differentiation and unlawful discrimination.

## 2.4 Statistical Disparities and *Prima Facie* Discrimination

To initiate a claim for discrimination, a claimant must establish a *prima facie* case [37, 40, s 136]. Sufficient evidence must be produced to show that unlawful discrimination may have occurred, including by showing discriminatory effects or harm against an individual or group caused by the decision-maker's action [37, 65]. Statistical evidence can be used to prove less favourable treatment or particular disadvantage, but by design, it shows correlations, and "a correlation is not the same as a causal link" [130, para. 28]. We explain the threshold for legal causation at the trial stage in Section 2.5. Although, at this stage, a mere correlation between the adverse effect on the person and the decision-maker's action will suffice [65]. The size of the disparity is relevant. Smaller disparities are less likely to trigger legal inquiry under anti-discrimination laws [127]. Courts will compare statistical evidence showing the different effects and outcomes between a disadvantaged group compared to a group without the protected attribute. The significance of the statistical disparity hinges on the specifics of the case [127, 124]. The thresholds for statistical significance are flexible and often resisted by courts to avoid excessive dependence on data [138]. The UK has specifically avoided thresholds like those used to measure statistically significant disparity in the US [105, 10].

Statistical disparities, as identified through algorithmic fairness metrics, may indicate a reason to consider whether discrimination has arisen. However, without taking context and potential true and

legitimate differences into account, these disparities hold little legal weight (see Section 2.1). We can
formalise this as the legal target being to minimise the conditional estimation disparity

$$\omega = ||\mathbb{E}_x\left[\epsilon_i \mid x_l, x_p\right] - \mathbb{E}_x\left[\epsilon_i \mid x_l\right]||_2, \tag{6}$$

where $|| \cdot ||_2$ is the euclidean norm. This target generalises the idea of minimising conditional
statistical parity. If we assume *true* conditional statistical parity, i.e.

$$\mathbb{E}_x\left[p(y_i|x_i^{\text{true}}) \mid x_l, x_p\right] = \mathbb{E}_x\left[p(y_i|x_i^{\text{true}}) \mid x_l\right], \tag{7}$$

then the target in Eq. 6 will be reduced to minimise the conditional statistical parity (see Eq. 3).
Although, this is only true as long as there are no true differences.

Hence, if true statistical parity does not hold, it is explained by true differences between groups. If
there is a true difference, such as age in financial services, forcing conditional statistical parity would
harm the protected group, most likely resulting in unlawful discrimination. This result aligns with
previous observations about the risks of forcing parity metrics [31, 144, 54]. Courts may need to be
more flexible in the type of statistical data they consider to establish a *prima facie* case by considering
non-comparative adverse effects in their assessment. Therefore, deferring to conditional estimation
parity provides an avenue for a contextually informed assessment.

## 2.5 Legal Causation and the Utility Function

To lawyers, causation is the relationship between an act, i.e., an action or decision, and its effect,
which requires two questions: (1) factually, *but for* the act, would the consequences have occurred;
(2) is the act a substantial cause of the consequence to apply responsibility. We are concerned with
the first question. Direct discrimination "requires a causal link between the less favourable treatment
and the protected characteristic"; indirect discrimination "requires a causal link between the PCP and
the particular disadvantage suffered by the group and individual" [130, para. 25]. In an algorithmic
context, this causal link requires asking whether $i$ would have received the same action or decision
$a$, *but for* their protected attribute $x_p$ or the PCP that indirectly relates to their protected attribute
$x_p$ [122, 123]. For instance, whether an individual would have suffered the disadvantage *but for* the
protected attribute would be discriminatory regardless of the decision-maker's intention [1]. This is a
notable distinction from certain aspects of US discrimination doctrine.

From a decision-theoretic perspective, the protected attribute $x_p$ can affect the decision $a$ either
through the utility function $u(a, y)$ or through the model $\hat{p}(y|x)$. Discrimination may occur if
the utility function in Eq. 1 differs for different groups defined by the protected attribute. Such a
difference would mean that an individual or whole group with a protected attribute is treated less
favourably than those without a protected attribute given the same model $\hat{p}(y|x)$. Such a difference in
the utility function would risk unlawful discrimination. Specifically, if $u(a, y)$ is changed for different
persons, either *directly* based on a protected attribute or *indirectly* has the effect of disproportionately
disadvantaging a group with a protected characteristic without justification (see Section 2.6).

Having different $\hat{p}(y|x)$, on the other hand, would mean that there is a legal causation between the
decision $a$ and $x_p$. This might either be motivated by true differences (see Section 2.1) or a result of
conditional estimation disparity. In the latter case, this might be a case of legal causation, i.e., that the
model is poor, and hence, the modelling has resulted in disadvantaging a protected group. Therefore,
we can view the causal structure of $\hat{p}(y|x)$ as central to avoiding unlawful discrimination. However,
not considering causal structures could lead to conditional estimation disparity, and potentially result
in unlawful discrimination.

Legal causation focuses on the legal causal link between $x_p$ and the decision $a$. In addition, legal
causation is less formal than common definitions of causal effects in ML. Courts, at least outside of
the US, are effects-orientated, and a wide range of forms of a "legal causal link" could be identified
[109, 65]. Much of the causal-based fairness literature formulates "causation" on the true causal
model structure in $\hat{p}(y|x)$, i.e., the study of the causal effect of $x$, due to outside interventions on $y$
[101, 10, 149, 21]. However, this formulation is not the same as that of legal causation.

In this discussion, the parallels to other discrimination studies become evident in how it would
affect automated decision-making, particularly taste-based and statistical discrimination. Taste-based
discrimination[11], could arise if only the utility function $u(a, y)$ unjustifiably disfavours a group
based on protected attributes $x_p$. Statistical discrimination, on the other hand, arises when decision-
makers use group-level statistics as proxies for individual characteristics due to imperfect information

[7, 102]. Statistical discrimination parallels the disadvantaging of a group due to having different $\hat{p}(x|y)$. While these types of discrimination are generally prohibited, statistical discrimination can be legally permissible in some circumstances (see Section 2.1).

## 2.6  Legitimate aim and $y$

Decision-makers must consider the legitimacy of using an SML model by explicitly defining its purpose and the outcome variable $y$. In algorithm design, social implications should be considered [87, 57, 63]. Additionally, this aligns the model's use with legal expectations.

If the court believes sufficient evidence of discrimination exists, the burden shifts to the respondent to disprove allegations of unlawful discrimination [38]. Indirect discrimination can be justified if the PCP is a proportionate means of achieving a legitimate aim [40, s 19(2)(d)]. Identifying a legitimate aim is closely connected to the choice of $y$, the unknown entity used for decision-making. If the choice of $y$ is legitimate based on context and the benefit outweighs any potential harm, there is a lower risk of unlawful discrimination [35].

The legitimacy of the aim depends on the decision-makers' *raison d'être* [65]. In *Homer*, the Court established a legitimate aim must "correspond to a real need and the means used must be appropriate with a view to achieving the objective and be necessary to that end" [128, 35, 39]. In lending, it is a legitimate aim to protect the repayment of their loans or at least secure their loans. In fact, "the mortgage market could not survive without that aim being realised" [126, para. 79].

For a legitimate $y$ to be an exception to indirect discrimination, the PCP must be a proportionate means of achieving the legitimate $y$ [40, s 19(2)(d)]. To be proportionate, it must be an appropriate means of achieving the legitimate aim and (reasonably) necessary to do so [128]. Such analysis will turn on the facts of each case. However, it will require evaluating whether the design choices were "appropriate with a view to achieving the objective and be necessary" by weighing the need against the seriousness of detriment to the disadvantaged group [39, para. 151]. This will require considering whether non-discriminatory alternatives were available [128]. Measures to improve accuracy, maximise benefits over costs, minimise estimation error, or condition for protected attributes may all be relevant considerations for whether the modeller's choices were proportionate means of achieving a legitimate $y$.

If the estimated outcome $\tilde{y}$ approximates the true outcome $y$, this can lead to biased predictions. Let

$$\gamma_i = ||p(\tilde{y}_i|x_i^{\text{true}}) - p(y_i|x_i^{\text{true}})||_2 \, , \tag{8}$$

then, if the expectation of $\gamma$ condition on $x_l$ shows a disparity, i.e.,

$$\mathbb{E}_x[\gamma \mid x_p, x_l] \neq \mathbb{E}_x[\gamma \mid x_l] \, , \tag{9}$$

it suggests the use of $\tilde{y}$ is inappropriate and might be discriminatory.

To illustrate with an example, if a bank's training data is outdated or sourced from a different country, it may not accurately represent the current population relevant to the model. This discrepancy can lead to biased estimations, particularly if the data reflects historical prejudices. For instance, the model might unjustly associate certain demographics with higher default risk, not because of true differences but biased historical data [as warned in 33].

## 2.7  Legitimate $x$

One of the more crucial aspects of SML for automated decision-making is the choice of features $x$. The aim and $y$ will help inform the choice of features to include in the model. We can separate three types of features from a legal perspective: features with protected attributes $x_p$, legitimate features $x_l$, and non-legitimate or illegitimate features $x_n$. The distinction between $x_l$ and $x_n$ depends on whether the feature can be considered legitimately related to $y$ (see Section 2.5). Causal fairness literature has engaged with questions of discriminatory variables through the lens of proxy discrimination [69, 115]. Proxy discrimination has a specific legal meaning under UK law that relates to direct discrimination, unlike much of the US literature on proxy discrimination that relates to indirect forms of discrimination. Here, we explain the UK legal implications of such causal relationships between variables and we provide a real-world example in Appendix A.

### 2.7.1 Direct Discrimination and Removing $x_p$

Direct discrimination in automated decisions may arise when members of, or an entire protected group, is affected. Where a model $\hat{p}(y|x)$ uses a protected attribute $x_p$, and there is a difference in predictions between the protected groups defined by $x_p$, this risk arises. Models have directly used protected characteristics, giving rise to direct discrimination [94, see discussion in Appendix]. Direct discrimination may arise when a feature is an exact proxy for a protected attribute. In *Lee v Ashers*, Lady Hale explained that the risk of direct discrimination also arises if a decision is based on a feature that "is not the protected characteristic itself but some proxy for it" [129]. Therefore, direct discrimination can arise even where $x_p$ has been removed because there is a feature which is an *exact proxy* that is "indissociable" or has an "exact correspondence" to $x_p$ [130, 129]. Formally, we can define an exact proxy as a feature $\tilde{x}_p$ with a perfect or almost perfect correlation with $x_p$ [115].

UK courts have accepted that an exact proxy would be pregnancy because "pregnancy is unique to the female sex" [121, 125]. If a model uses pregnancy or maternity leave as a feature, collected from CV information, for example, it would have the effect of using an exact proxy $\tilde{x}_p$ that could hypothetically be the basis for a direct discrimination claim.

Given the relevance of $x_p$ to direct discrimination, modellers have been encouraged to remove protected attributes when designing ML models [105, 62, 48]. These claims are usually based on the US Equal Protection Clause, which subjects classifications based on certain protected characteristics, such as race, to strict scrutiny [146]. The focus on excluding certain data inputs is one form of discrimination prevention [146, 46], but not under UK law. Further, simply removing protected characteristics reduces accuracy and utility [150, 66], and does not remove the risk of discrimination [34, 79, 72].

This reasoning connects to the true DGP. If a protected attribute like gender is inherent in the DGP, removing it does not eliminate discrimination but instead may introduce it. Taking a gender-neutral approach to recidivism predictions may have the adverse effect of discrimination against women who would otherwise have received lower risk scores [31]. In *Loomis*, the Court accepted that in recidivism algorithms, "if the inclusion of gender promotes the accuracy, it serves the interests of institutions and defendants, rather than a discriminatory purpose" [112, 766]. Hence, if the inclusion of $x_p$ improves the accuracy and benefits the protected group, it may avoid the risk of discriminatory purposes. There is an absence of any legal guidance in the UK on the relationship between true probabilities and protected attributes in automated decision-making. Pending further legal guidance, it is important to carefully consider whether including $x_p$ is relevant to promote accuracy and conditional estimation parity. Removing protected attributes often ignores the true probabilities for the legitimate differences between protected groups, affecting the lawfulness of its outcomes.

Therefore, removing $x_p$ will not avoid liability for unlawful direct discrimination by itself. Even if a model ignores $x_p$, in practice, it may rely on other data points acting as proxies with "exact correspondence" to a protected characteristic $\tilde{x}_p$. Importantly, this diverges from US law and highlights that intention is immaterial to UK direct discrimination [Cf. e.g., 5, 115]. UK law focuses on the discriminatory effects rather than a formalistic view of whether $x_p$ is considered or not.

### 2.7.2 Defining $x_l$ and $x_n$

Indirect discrimination may arise if a PCP appears to apply equally to everyone but disadvantages members of a protected group. Both forms of discrimination can arise using an exact proxy or a weak proxy in a PCP. Therefore, identifying legitimate features is challenging when many features correlate to protected groups. We define non-legitimate features $x_n$ as features not legitimate in the context of the true DGP (Section 2.2). In practice, this means a non-legitimate feature is one that, if included, would not contribute to the predictive performance of the optimal model, i.e., the one with the lowest estimation error (Section 2.3). Therefore, $x_n$ would not improve the predictive performance if a modeller had the true features.

For example, hair length strongly correlates to gender in many cultural contexts but is unlikely to contribute to the consumers' true default risk. Boyarskaya et al. explain the absence of a "causal story" between hair length and loan repayment because hair length would not be part of a true model for the risk of default [19]. Therefore, hair length is an example of $x_n$ in a lending context.

For comparison, the legitimacy of zip codes illustrates the nuanced nature of legitimate features. While a zip code may correlate with race in some contexts, it might be a legitimate variable in other situations. For example, in an application for home insurance covering flood risk, zip codes

are invaluable proxies for granular information such as geographical features, land topography and historical flooding. Therefore, in the best model for property flood insurance decisions, zip code will improve the predictive performance as a legitimate proxy for data within the true DGP. However, in a university application, there should be no predictive or causal relationship to merit for acceptance. In such cases, zip code likely acts as a proxy for race or the unprotected characteristic of socio-economic status and would be $x_n$. So, in some circumstances, the zip code would be legitimate $x_l$, but in others, it may not be $x_n$. It will also be relevant to consider whether a less discriminatory feature is available, i.e., one with less correlation to a protected attribute that is equally predictive.

As explored in Appendix A, in lending, information about income and debts are likely to be legitimate features $x_l$. Credit scores can be a proxy for a person's financial position, as well as protected attributes [18, 60]. However, the complexity of calculating credit scores means it is more valuable for inferring income, debt repayments, and history of credit. Credit scores, or related features, would have a material impact on the true model for default, and then would be a legitimate feature $x_l$.

Given that nearly, all features may contain some information on protected attributes, even legitimate factors [30], this approach explains the need to assess the strength of this dependence and whether the feature contributes significantly to the model's prediction and can be argued to be part of a true DGP.

### 2.7.3  Feature construction from $x$

The distinction between $x_l$ and $x_n$ also gives rise to problems in automatic feature construction, such as using deep neural networks. If features are constructed automatically using a combination of $x_l$ and $x_n$, indirect and direct discrimination are risks. As an example, an applicant's resume contains legitimate features $x_l$ for recruitment prediction. However, the detailed granularity of many resumes also gives rise to the problem of non-legitimate information, such as maternity leave or women-only sports or other information that may contain information on other protected attributes. Hence, there needs to be an active choice of only including legitimate features $x_l$ from available data in the model.

## 3  Conclusions

Minimising unlawful discrimination in automated decision-making requires a nuanced and contextual approach. While it is beyond our scope to offer specific legal advice, our findings underscore several key considerations to identify and mitigate potential discrimination effectively:

1. *Assess data legitimacy*. Carefully examine if the data, both the target variable ($y$) and features ($x$), are legitimate for the specific context (Sections 2.6 and 2.7). Legal analysis should inform what is legitimate in a specific setting.
2. *Build an accurate model*. Strive to approximate the true DGP $p(y|x)$, using only legitimate features $x_l$. Reasonable, necessary, and proportionate steps must be taken to minimise estimation error and aim for estimation parity (Section 2.3). This may entail model inference, interrogating social biases in the data, and scrutinising the estimated model.
3. *Evaluate statistical disparity*. Given the best model $\hat{p}(y|x)$, assess for conditional statistical parity by examining outcomes across groups with protected characteristics (Section 2.4). If a model's performance improves by including protected attributes, consider:
   (a) Identify whether conditional statistical parity is unattainable or undesirable based on true group differences. This requires stringent analysis into whether differences stem from prior injustice or legitimate variation.
   (b) Incorporate further legitimate features $x_l$ that could minimise statistical disparities by "explaining away" the performance gained by the protected attribute with legitimate features.
   (c) Avoid using the model due to unmitigated discrimination risks.

While these guidelines cannot guarantee lawful automated decisions, they provide meaningful recommendations and abstractions to help identify and mitigate unlawful discrimination risks.

In conclusion, this work bridges a critical gap between the technical aspects of automated decisions and the complexities of anti-discrimination law. By translating these nuanced legal concepts into decision theory, we underscore the importance of accurately modelling true data-generating processes and the innovative concept of estimation parity. This interdisciplinary approach enhances the understanding of automated decision-making and sets a foundation for future research that aligns technological advancements with legal and ethical standards.

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

# A    Case Study

**Overview of Finnish Anti-Discrimination Law**

Finnish anti-discrimination law bears many similarities to UK and EU laws. We briefly set out the relevant provisions that show the similarities to the Equality Act set out in Section 1.3.

Section 8(1) of the Non-Discrimination [84] defines the protected characteristics as:[1]

> No one may be discriminated against on the basis of age, origin, nationality, language, religion, belief, opinion, political activity, trade union activity, family relationships, state of health, disability, sexual orientation or other personal characteristics. Discrimination is prohibited, regardless of whether it is based on a fact or assumption concerning the person him/herself or another.

Section 3(1) of the Non-Discrimination Act provides that: "Provisions on prohibition of discrimination based on gender and the promotion of gender equality are laid down in the Act on Equality between Women and Men (609/1986)." The Non-Discrimination Act can be applied in cases of multiple discrimination, even if gender is one of the grounds of discrimination [83, 84, s 3(1)].

It is worth noting that this definition is broader than in the UK Equality Act. Some protected characteristics are outlined more explicitly; for example, a person discriminated against on the basis of language may be able to bring a claim based on racial discrimination [120]. Unlike many Nordic countries, the Equality Act does not explicitly protect political activity, trade union activity, and does not include "or other personal characteristics" [53].

Direct discrimination is defined in Section 10:[2]

> Discrimination is direct if a person, on the grounds of personal characteristics, is treated less favourably than another person was treated, is treated or would be treated in a comparable situation.

Indirect discrimination is defined in Section 13:[3]

> Discrimination is indirect if an apparently neutral rule, criterion or practice puts a person at a disadvantage compared with others as on the grounds of personal characteristics, unless the rule, criterion or practice has a legitimate aim and the means for achieving the aim are appropriate and necessary.

Section 11(1) defines justifications for different treatment as:[4]

> Different treatment does not constitute discrimination if the treatment is based on legislation and it otherwise has an acceptable objective and the measures to attain the objective are proportionate.

**Overview of Finnish National Non-Discrimination and Equality Tribunal Decision 216/2017**

The first case regarding automated decision-making and discrimination was in Finland. The person, referred to as A, was denied credit for online purchases based on a credit rating system employed by a bank. Person A reported the case to the Non-Discrimination Ombudsman (Yhdenvertaisuus-valtuutettu), who brought the case before the National Non-Discrimination and Equality Tribunal (Yhdenvertaisuus- ja tasa-arvolautakunta). The Tribunal found that the bank's statistical scoring model resulted in direct discrimination based on multiple protected characteristics and was not

---

[1]Official translation from Finnish, although only legally binding in Swedish (not included) and Finnish: "Syrjinnän kielto Ketään ei saa syrjiä iän, alkuperän, kansalaisuuden, kielen, uskonnon, vakaumuksen, mielipiteen, poliittisen toiminnan, ammattiyhdistystoiminnan, perhesuhteiden, terveydentilan, vammaisuuden, seksuaalisen suuntautumisen tai muun henkilöön liittyvän syyn perusteella. Syrjintä on kielletty riippumatta siitä, perustuuko se henkilöä itseään vai jotakuta toista koskevaan tosiseikkaan tai oletukseen"

[2]"Syrjintä on välitöntä, jos jotakuta kohdellaan henkilöön liittyvän syyn perusteella epäsuotuisammin kuin jotakuta muuta on kohdeltu, kohdellaan tai kohdeltaisiin vertailukelpoisessa tilanteessa."

[3]"Syrjintä on välillistä, jos näennäisesti yhdenvertainen sääntö, peruste tai käytäntö saattaa jonkun muita epäedullisempaan asemaan henkilöön liittyvän syyn perusteella, paitsi jos säännöllä, perusteella tai käytännöllä on hyväksyttävä tavoite ja tavoitteen saavuttamiseksi käytetyt keinot ovat asianmukaisia ja tarpeellisia."

[4]"Erilainen kohtelu ei ole syrjintä, jos kohtelu perustuu lakiin ja sillä muutoin on hyväksyttävä tavoite ja keinot tavoitteen saavuttamiseksi ovat oikeasuhtaisia."

justified by an acceptable objective achieved by proportionate measures. Consequently, the Tribunal prohibited the bank from continuing this practice and imposed a conditional fine to enforce compliance.

The decision-making system in question is for online store financing, which is a purchase-bound, fast and automated credit type very different from regular consumer credit. The credit applied for by the consumer in each situation is also always bound to the purchase and its value, which means that it is more difficult, or even impossible, to undertake detailed requests for information and background checks. The individual investigation of the creditworthiness of customers using personal information and documents, such as salary and tax certificates, may not be suitable for this type of credit.

**Decision-making Model and Data**

The company made credit decisions based on data from the internal records of the credit company, information from the credit file, and the score from the company's internal scoring system.

The bank's scoring system assessed creditworthiness. The scoring system used population statistics and personal attributes to calculate the percentage of people in certain groups with bad credit history and awarded points proportionate to how common bad credit records were in the group in question.

The variables used included race, first language, age, and place of residence. The company did not require or investigate the applicant's income or financial situation.

**True Data Generating Process and Estimation Error**

The bank's scoring model was based on statistical correlations calculated population and groups, including gender, language, age and place of residence, meaning the model is more or less $\hat{p}(y|x_p)$.

This model cannot be said to have attempted to model the true underlying data-generating process and instead relied on data that was available regarding protected attributes. It is reasonable to expect that the bank was aware of other legitimate factors that could explain the credit score. Therefore, the model introduces epistemic uncertainty stemming from the lack of information that could have been used to make better predictions, i.e. reasonable legitimate features $x_l$.

By solely using the data available, rather than identifying what data would be best to reduce estimation error, the modellers built an automated decision-making system that unlawfully discriminated. We now evaluate how the Tribunal came to those conclusions about the legitimacy of $y$ and $x$ for such a model.

**Legitimate $y$**

The bank argued that the "different treatment does not constitute discrimination if the treatment is based on legislation and has an otherwise acceptable objective and the measures to attain the objective are proportionate." The Tribunal agreed that "the provision of credit to customers is a business, the purpose of which is to gain profit" and that "the investigation of creditworthiness is as such based on law and that it has the acceptable and justified objective as defined in section 11 of the Non-Discrimination Act". Therefore, creditworthiness assessment is a legitimate $y$.

However, the Tribunal clarified that "the individual assessment required by the legislation means expressly the assessment of an individual's credit behaviour, credit history, income level and assets, and not the extension of the impact of models formed on the basis of probability assessments created with statistical methods using the behaviour and characteristics of others, to the individual applying for the credit in the credit decision in such a way that assessment is solely based on such models." Therefore, to be appropriate and necessary to achieve that aim, the model must consider legitimate features $x_l$.

**Protected, Legitimate, and Non-Legitimate Variables $x$**

Four protected attributes were used as variables in this model $x_p$: age, language, other personal characteristic (place of residence), and gender.

The Tribunal acknowledged that age may be a legitimate variable if it had been used in the assessment of creditworthiness mainly when applied to young persons. However, it was not justified in this assessment, given the age of the credit applicant.

The Tribunal agreed with the position under European law that gender is prohibited from being used as an actuarial factor in financial services [42].

Therefore, these features did not contribute to the accuracy of the model's prediction in a way that could be argued as part of the true DGP. Therefore, in this case, these $x_p$ variables are also non-legitimate variables $x_n$.

As explained by the Tribunal, to achieve the legitimate $y$ of undertaking an individual assessment of creditworthiness and ability to repay, the model should have considered, for example, income, expenditure, debt, assets, security and guarantee liabilities, employment and type of employment contract (i.e., permanent or temporary). These features would have been legitimate variables $x_l$ by improving the predictive performance of the model to achieve more accurate decisions.

**Conditional Estimation Parity**

Using the legitimate variables identified above, we can now consider conditional estimation parity, the difference in estimation error between groups with a protected attribute, given legitimate features. Reducing the error in Eq. 4 is expected to diminish the risk of conditional estimation disparity. However, assessing conditional estimation parity is complex due to inherent challenges in evaluating estimation error.

Judges engage this type of reasoning through statistical or theoretical means. In this case, the Ombudsman brought evidence of the effects of the protected characteristics $x_p$ on the true prediction.

Person A was negatively affected by his **age**. He was in the age group of 31-40 years old, but if he had been at least 51 years old, he would have received a higher score sufficient for the credit application.

If person A spoke Swedish as his first **language**, he would have received a sufficient score for granting the loan. Finnish-speaking residents received a lower score compared to Swedish-speaking residents. Further, ethnic minorities with an official first language other than Finnish or Swedish were put in an unfavourable position.

A would have earned more points based on his **residential area** if he had lived in a population centre. The bank's statistical method, which is based on a grid of residential areas, gave A the lowest score because he lives in a sparsely populated area that has not yielded any statistically significant information.

**Gender** impacted the model, where women received a higher score than men. The Tribunal agreed that if the person A had been a woman, he would have been granted the credit.

**Conclusions**

This case study demonstrates the intersection between judicial reasoning and our formalisation. To avoid liability for unlawful multiple direct discrimination in this algorithmic decision-making process, the company should have:

1. *Assessed data legitimacy*. While the Tribunal agreed with the target variable ($y$) as a legitimate aim, they did not believe the features ($x$) were legitimate for the specific context (Section 2.7).
2. *Built an accurate model*. The bank did not strive to approximate the true DGP $p(y|x)$, and did not use legitimate features $x_l$. Reasonable, necessary, and proportionate steps should have been taken to minimise estimation error and aim for estimation parity (Section 2.3).
3. *Evaluate differences*. The bank should have considered whether there were true and legitimate differences based on protected characteristics and whether they could have been "explained away" by legitimate features ($x_l$) to minimise statistical disparities.

These recommendations should be used to help identify and mitigate unlawful discrimination within the specific context of each jurisdiction.

