# OpenReview forum: "Formalising Anti-Discrimination Law in Automated Decision Systems"
_NeurIPS.cc/2024/Conference — Submitted to NeurIPS 2024_

### Official Review · Reviewer_e8Fx · 2024-06-25

**Soundness:** 4
**Presentation:** 4
**Contribution:** 3
**Rating:** 8
**Confidence:** 3

**Summary:**

The paper presents a formalization of fairness metrics intended to ease analysis of discrimination by automated decision making systems in the UK. While there is a relatively applied angle, the bulk of the contribution is intended to be a generic and re-targetable mathematical formalism.

**Strengths:**

This paper shows significant strength in its understanding of nuance with the way law works–something that is sorely missing from the vast majority of CS papers that attempt to handle legal concepts. I was very pleased overall by the mapping the authors performed between relevant legal concepts in the UK and their formal model of fairness. The bulk of the contribution here is in the modelling–which while it results in a simple formulation, should not be taken to undercut the value of the contribution.

Non-US legal contexts often get left out of the literature, even common law jurisdictions–yet they impact a significant number of people, and this work takes formalising fairness across that rubicon.

**Weaknesses:**

I do not have any major scientific critiques, though there were areas where the clarity of the paper could improve.

Lines 240-274 were written in harder to parse prose than the bulk of the rest of the paper. I had to reread that area multiple times.

The case study in Appendix A was actually very useful for understanding the authors' formalism and it is a shape that some of that context was not woven into the paper as concrete examples of how to understand the math.

The discussion on proxy discrimination never seemed to finish? I wasn't able to understand its meaning under UK law.


Missing a ref to Homer on L299.

All these are very minor issues. I'm substantially in favour of accepting this paper.

**Questions:**

Where does proxy discrimination sit under UK law?

On L428 the authors make the suggestion to incorporate legitimate features that substantively create the same outcome as the protected features–this seems like a recipe for disaster in a world where we care about the ultimate effects? I'm wondering how we can square the circle there!

**Limitations:**

Ultimately, adherence to a formalism is *not* what courts generally take into account. While statistical analyses may be used to advance a given line of argument, the standards used are open-textured–and this is an inherent limitation of this line of work.
It also would have been good to see where this formalism sits under EU law (or representative EU-member law) or perhaps a discussion of how civil law jurisdictions handle these sorts of issues.

---

> ### Author Rebuttal · Authors · 2024-08-06
>
> Thank you for your thoughtful and highly positive feedback. We appreciate your recognition of our work’s **"significant strength in its understanding of nuance with the way law works"** and that you were **"very pleased overall by the mapping the authors performed between relevant legal concepts in the UK and their formal model of fairness"**, and our work on non-US contexts **"takes formalising fairness across that rubicon"**.
>
> We especially appreciate **“The bulk of the contribution here is in the modelling–which while it results in a simple formulation, should not be taken to undercut the value of the contribution.”**
>
> ## Weaknesses
>
> > **W1** “Lines 240-274 were written in harder to parse prose than the bulk of the rest of the paper.”
>
> **R1** We will revise these paragraphs to improve clarity and readability.
>
> > **W2** “The case study in Appendix A was actually very useful for understanding the authors' formalism and it is a shape that some of that context was not woven into the paper as concrete examples of how to understand the math.”
>
> **R2** We appreciate your feedback on the usefulness of the case study in Appendix A. We agree that incorporating elements of this case study into the main text would enhance the paper's clarity. We will revise the paper to incorporate relevant examples throughout the main text, particularly to provide illustrations of the formalism and connect the main text more to the appendix throughout.
>
> > **W3** “The discussion on proxy discrimination never seemed to finish? I wasn't able to understand its meaning under UK law.”
>
> **R3** We apologise for the lack of clarity in our discussion of proxy discrimination. Under UK law, proxy discrimination is primarily considered in the context of direct discrimination. It occurs when a decision is based on a criterion that is not explicitly a protected characteristic but is so closely associated with it as to be indistinguishable.
>
> For example, in _James v Eastleigh Borough Council_, using the state pension age as a criterion for free swimming pool entry amounted to direct sex discrimination, as the state pension age was different for men and women at the time. Similarly, pregnancy has been held as a proxy for sex discrimination (_O’Neil_; _Webb_).
>
> The literature on proxy discrimination, particularly in the US domain, focuses on the concept of disparate treatment (indirect discrimination), where the use of a protected variable results in unequal outcomes for a particular group. Indirect discrimination may still arise in similar situations; however, the analysis is outcome-based, not input-focused. Therefore, indirect discrimination through the use of proxy variables will be examined by whether a group of people with a protected attribute is put at a particular disadvantage when compared to those without such attributes (Equality Act s 19(2)).
>
> We will clarify and improve the discussion of proxy discrimination in Section 2.7.
>
> > **W4** “Missing a ref to Homer on L299.”
>
> **R4** We will move the reference to _Homer_ up to the case name on L299 (it is currently at the end of the quote on L301, [128]). Thank you for catching this oversight.
>
> ## Questions
>
> > **Q1** “Where does proxy discrimination sit under UK law?”
>
> **A1** Please see comment in **R3** above.
>
> > **Q2** “On L428 the authors make the suggestion to incorporate legitimate features that substantively create the same outcome as the protected features–this seems like a recipe for disaster in a world where we care about the ultimate effects? I'm wondering how we can square the circle there!”
>
> **A2** Our suggestion at L428 is not to incorporate features that create the same outcome as protected features. Rather, we recommend incorporating legitimate features that could explain the apparent predictive power of correlated protected attributes in a non-discriminatory way. For example, if gender appears to be predictive of loan default, incorporating features like income stability or credit history (legitimate variables for loan decisions) may reduce or eliminate the apparent predictive power of gender due to lower financial base rates for women. This approach aims to capture true causal factors influencing the outcome rather than relying on protected attributes or their proxies. Does this address your question?
>
> ## Limitations
>
> We agree that it would be helpful to consider this formalism “under EU law (or representative EU-member law) or perhaps a discussion of how civil law jurisdictions handle these sorts of issues.”
>
> While not the case for all areas, UK anti-discrimination law is very similar to EU law. In fact, Lady Hale has explained that: “Much, but by no means all, of the Equality Act 2010 is derived from our obligations under European Union law. Those parts which are so derived must be interpreted consistently with EU law (as it is now called) and it is inconceivable that Parliament intended the same concepts to be interpreted differently in different contexts.” (_Essop_ [19]). Additionally, in Appendix A, the Finnish example illustrates a representative EU-member state that implements EU Directives which are substantially similar to UK anti-discrimination laws.
>
> We will outline these similarities explicitly in the paper and limitations section.
>
> It would indeed be an interesting avenue for future work to explore the differences in how civil law jurisdictions would consider these formalisms and their approach to algorithmic discrimination. We would expect formalisations to explicitly follow statutory definitions from civil codes and be designed to fit into a deductive framework for judges to investigate independently. We will consider comparable jurisdictions to expand this analysis in future research.
> ## Summary
> We are grateful for your thorough review and constructive feedback. We will address all points raised to enhance the clarity and impact of our paper. We appreciate your strong support and are committed to strengthening the paper further.

---

> > ### Comment · Reviewer_e8Fx · 2024-08-08
> > **Response to Rebuttal**
> >
> > Thanks for the detailed rebuttal.
> >
> > I still maintain my positive score.
> >
> > Regarding A2, if you could really work on clarifying that in the paper, I think it would go a long way to assuaging my concerns.
> >
> > Overall though, nice work!

---

### Official Review · Reviewer_Z976 · 2024-06-29

**Soundness:** 3
**Presentation:** 4
**Contribution:** 3
**Rating:** 7
**Confidence:** 1

**Summary:**

The paper maps existing literature and law on algorithmic fairness onto a decision-theoretic framework. It describes various desiderata (e.g. statistical parity) and legal restrictions (e.g., legitimate aims) in terms of expectations, distributions, estimation error, etc.

**Strengths:**

The paper is well-written and survey a large literature. It appears to state legal tests (particularly under U.K.) with care, while being careful not to overclaim about what its definitions actually establish.

**Weaknesses:**

n/a

**Questions:**

I regret to say that my expertise does not extend to the paper's two principal areas (anti-discrimination law and decision theory). I cannot form a sufficiently educated opinion about the correctness or the novelty of the paper's results. This is my fault, not the authors'.

---

> ### Author Rebuttal · Authors · 2024-08-06
>
> Thank you for your valuable feedback and for recognising the potential impact of our work.
>
> We're pleased you found our paper well-written, comprehensive in its literature survey, and careful in stating legal tests. We are encouraged by your high rating and assessment of our paper's potential impact on multiple areas.
>
> We appreciate your candid assessment regarding expertise limitations and thank you for your time. Please let us know if you have any further questions or suggestions during the rebuttal period.

---

### Official Review · Reviewer_4EE7 · 2024-07-07

**Soundness:** 2
**Presentation:** 2
**Contribution:** 1
**Rating:** 3
**Confidence:** 2

**Summary:**

- There is a gap between the definitions of fairness studied in the computer science literature, and the definitions of fairness operationalized by courts adjudicating discrimination claims. This limits the usefulness of the CS definitions.
- Amongst work attempting to reconcile legal and computational definitions of fairness, little has focused on anti-discrimination law outside the US.
- This paper makes four contributions in this context:
    - (1) It formalizes elements of anti-discrimination law into a decision-theoretic formalism
    - (2) If analyzes the legal role of the data-generation process
    - (3) It proposes conditional estimation parity as a legally-informed target
    - (4) It provides recommendations on creating SML models that minimize the risk of unlawful discrimination in automated decision-making

**Strengths:**

- The paper’s focus is interesting–the fairness literature is biased towards the US, and I imagine most fairness researchers would be unaware of subtle differences between UK and US anti-discrimination law.
- Because UK law is influential around the world, understanding how it regulates fairness in algorithmic systems has global importance.

**Weaknesses:**

- Much of the paper reads like a review of anti-discrimination law. This makes it difficult to parse out (1) what the technical contributions are, (2) why they’re novel, and (3) why they matter.
- It’s extremely unclear what the technical payoff of the paper’s modeling choices are. The fairness field is overwhelmed with different definitions/frameworks. Why is the one proposed by the author’s meaningful over others?
- It seems like an essential point to the paper’s argument is that prior work hasn’t studied UK anti-discrimination law. But if the paper wants to successfully extend that into an argument about modeling choices, I think it needs to explain why the existing definitions of fairness do not work for UK law.
- The recommendations provided are extremely general. Are these new or different from the many recommendations that already exist in the fairness/responsible AI literature?

**Questions:**

The comments in the weaknesses section list the relevant questions!

---

> ### Author Rebuttal · Authors · 2024-08-06
>
> Thank you for your comments and questions on our paper. We are encouraged that you found the paper’s focus interesting and acknowledged that **"the fairness literature is biased towards the US, and I imagine most fairness researchers would be unaware of subtle differences between UK and US anti-discrimination law."** Additionally, we appreciate your recognition of the global importance of this work, as **"UK law is influential around the world."**
>
> ## Weaknesses
>
> > **W1** “Much of the paper reads like a review of anti-discrimination law. This makes it difficult to parse out (1) what the technical contributions are, (2) why they’re novel, and (3) why they matter.”
>
> **R1**  Our work is the first to translate UK anti-discrimination law into a formal ML framework. In our response to Reviewer iALf at **R1**, we outline our contributions in depth, including ways we plan to make them more prominent.
>
> The novelty lies in bridging the gap between legal concepts and machine learning practices, which currently needs to be improved in the field. We believe that this is summarised well by the description of the NeurIPS Workshop on Regulatable Machine Learning (https://regulatableml.github.io/):
>
> “[...] there appears to be a considerable gap between current machine learning research and these regulatory policies. Translating these policies into algorithmic implementations is highly non-trivial, and there may be inherent tensions between different regulatory principles.”
>
> This paper contributes to the specific intersection between regulations and machine learning. This matters because it provides a more legally robust approach to fairness in ML, potentially reducing the risk of unlawful discrimination in automated decision-making systems.
>
> > **W2** “The fairness field is overwhelmed with different definitions/frameworks. Why is the one proposed by the author’s meaningful over others?”
>
> **R2** Our approach provides a comprehensive and legally relevant framework for evaluating unlawful discrimination in automated decision-making systems under UK law. Unlike most fairness metrics, our framework directly incorporates legal principles from UK anti-discrimination law (substantively similar to the EU, Commonwealth and other common law jurisdictions) into a decision-theoretic model, allowing for a legally grounded approach to fairness in machine learning, unlike most previous metrics.
>
> > **W3** “I think it needs to explain why the existing definitions of fairness do not work for UK law.”
>
> **R3**  Existing fairness metrics primarily focus on statistical disparity between groups, oversimplifying the nuanced approach required by UK anti-discrimination law. They often fail to account for legitimate differences in outcomes, causal relationships, and legal justifications for differential treatment that are recognised under UK law. Furthermore, most fairness metrics do not adequately address the concepts of indirect discrimination, estimation error, and the role of the DGP, all crucial considerations in evaluating unlawful discrimination under the UK legal framework.
>
> > **W4** “The recommendations provided are extremely general. Are these new or different from the many recommendations that already exist in the fairness/responsible AI literature?”
>
> **R4** Our approach differs from many existing recommendations in the fairness/responsible AI literature in several key ways:
> 1. Legal Foundation: Unlike many recommendations in the fairness literature, ours are explicitly derived from UK legal principles (which are more relevant to jurisdictions including the EU, commonwealth and other common law countries than US literature). For example, our emphasis on assessing data legitimacy is directly tied to UK legal concepts of "legitimate aim" and "proportionate means."
> 2. Rejecting Forced Parity: We diverge from approaches that recommend forcing statistical parity, which can sometimes result in actual discrimination. Our framework allows for justified differences between groups when they stem from legitimate factors, aligning with legal standards.
> 3. Focus on Legitimate Variables: We provide a clear framework for identifying and using only legitimate variables based on their causal relationship to the outcome of interest. This approach is more nuanced than blanket recommendations to remove all correlated features.
> 4. Acknowledgment of Model Limitations: We explicitly state that in some cases, it may not be legally permissible to use a model at all if discriminatory effects cannot be mitigated. This frank acknowledgment of potential limitations is often missing from more optimistic recommendations in the field.
> 5. Practical Relevance: As the case study in Appendix A demonstrates, these recommendations, while seemingly straightforward, are often not followed in real-world automated decision systems. This underscores the need for clear, legally-grounded guidelines.
> While some aspects of our recommendations may seem like common sense, their importance lies in their comprehensive nature and their alignment with legal standards. They provide a cohesive framework for developing fair ML systems that can withstand legal scrutiny under UK anti-discrimination law.
> ## Summary
> We appreciate your critical feedback and hope these clarifications address your concerns and more effectively communicate the novelty and importance of our work. Given these explanations and planned improvements, would you be willing to reconsider your score?

---

> > ### Comment · Reviewer_4EE7 · 2024-08-12
> >
> > Thank you–I really appreciate the response. I will stick with my score however.

---

### Official Review · Reviewer_XoDZ · 2024-07-10

**Soundness:** 3
**Presentation:** 3
**Contribution:** 3
**Rating:** 6
**Confidence:** 4

**Summary:**

This paper addresses the issues around existing fairness metrics and bias detection/mitigation methods not corresponding with legal notions of fairness, specifically under UK anti-discrimination law. The authors propose a theoretical framework for a data-generating process that aims to formalise the legitimacy of decisions and features in the data. Further, they propose a new metric "conditional estimation parity" which compares estimation errors for different protected groups.

**Strengths:**

1. The paper is well written and coherent. It translates potentially inaccessible legal scholarship and discussions clearly for a technical audience.
2. There is interesting discussion and the paper combines existing literature well. Although these discussions are not particularly novel, UK Equality Law in particular is rarely discussed and the investigations done here are useful to extend the literature for this niche.
3. The work addresses some big limitations in existing literature such as existing fairness metrics not aligning with legal notions of discrimination, particularly under non-US regulations, not considering context of what features are legitimate for an application or considering the estimation errors of decisions.

**Weaknesses:**

1. A lot of the paper is background or a collation of existing literature. The main contribution is the new conditional estimation metric metric but this metric relies on the true DGP and evaluating the estimation error which, as stated, can be complex in practice. This could make it difficult to use the metric in practice.
2. I understand it would be hard to use the metric for evaluating discrimination in existing datasets for the reasons specified above and also due to the inherent context-dependency of the metric (which is a benefit) but it could be useful to include some experimentation or results in a hypothetical scenario to show how it might be used in practice. As there are no results as such to comment on, it is difficult to assess it's significance.
3. The conclusions drawn such as "Assess data legitimacy" or "Build an accurate model", although justified with evidence in the paper, are not novel and are pretty standard, common-sense recommendations.
4. Overall, the main novel contribution is the new metric but this is a small part of the paper. The rest of the paper is a nice collation and narrative of existing literature but I am not sure it significantly advances the field.

Other comments:
1. I can't see where SML terminology is introduced - I assume this means supervised machine learning?
2. In Section 1.4, DGPs are mentioned for the first time. It would be useful to have some more background to them before this - what exactly is a DGP? I do not believe it is ever explained.

**Questions:**

1. See weaknesses above. How would you go about using the metric in a real scenario?
2. Do you have any thoughts about how your metric relates to other notions of fairness such as individual fairness metrics?
3. Could you explain "taste-based discrimination" further?

**Limitations:**

The authors are honest about the strengths and weaknesses of their work (although some are hidden away and not pointed towards in the checklist). It would be useful to improve the discussion of limitations in Section 1.4 as it only mentions the limitation of applicability only in the UK.

---

> ### Author Rebuttal · Authors · 2024-08-06
>
> Thank you for your thoughtful and constructive feedback. We appreciate that you found our paper **"translates potentially inaccessible legal scholarship and discussions clearly for a technical audience"** and **“addresses some big limitations in existing literature”**.
>
> ## Weaknesses
>
> > **W1** “The main contribution is the new conditional estimation metric…which, as stated, can be complex in practice.”
>
> **R1** We acknowledge that the metric relies on the true Data Generating Process (DGP) and evaluating estimation error, which can be complex in practice. However, our paper adopts a platonistic view of the true DGP, treating it as a conceptual tool rather than a concrete object; similar to how courts might reason about an idealised decision-making process. The DGP is well-established in model inference literature (e.g., Akaike, 1973; White, 1982; Vehtari & Ojanen, 2012). By connecting fairness and legal reasoning to a DGP, we enable the use of classical model inference tools, facilitating the operationalisation of these metrics by other researchers.
>
> > **W2** You suggest “experimentation or results in a hypothetical scenario to show how it might be used in practice.”
>
> **R2** While we understand the value of applied work, the primary purpose of this paper is to introduce a new theoretical framework that does not currently exist in the literature. It is, therefore, beyond the scope of one paper to introduce a new theoretical framework and simultaneously operationalise it. We will use our existing case study throughout the text to more clearly illustrate how our framework relates to courts’ legal reasoning in practice.
>
> > **W3** “The conclusions drawn such as ‘Assess data legitimacy’ or ‘Build an accurate model’, although justified with evidence in the paper, are not novel and are pretty standard, common-sense recommendations.”
>
> **R3** We agree that the recommendations may seem like common sense. However, some fairness literature recommendations contradict ours, such as advocating for statistical parity constraints or overlooking legal situations where model use may be prohibited. Also, the court case in Appendix A shows that, in practice, this is not common sense in practical automated decision systems.
>
> In addition, while some of our recommendations may seem common-sense, their importance lies in their grounding in legal principles and their specific application to fairness in machine learning.
>
> > **W4** “Overall, the main novel contribution is the new metric but this is a small part of the paper. The rest of the paper is a nice collation and narrative of existing literature but I am not sure it significantly advances the field.”
>
> **R4** Our contribution extends beyond the new metric and literature collation (see **R1** for Reviewer iALf). By connecting fairness metrics to actual legal reasoning, we advance fairness considerations to include legal limitations and reasoning, crucial for real-world applications. Our paper introduces new fairness principles, formalises a decision-theoretic approach to UK anti-discrimination law, and provides detailed analysis for modellers to use when developing ML systems. We will revise the introduction to highlight our contributions more explicitly and earlier in the paper.
>
> ## Other comments
>
> We will introduce SML as supervised machine learning and expand our explanation of the DGP. A DGP refers to the true, underlying process that produces the data we observe. It encompasses all factors and relationships that determine how data points are created--it is the “true” model that, if known, would perfectly describe how the data came to be and how future data would be generated.
>
> ## Questions
>
> > **Q1** “How would you go about using the metric in a real scenario?”
>
> **A1** While this paper doesn't operationalize the metrics in a real-world scenario, the idea of estimating a model given a DGP has a long tradition in model inference, e.g., using cross-validation and information criteria. Detailed operationalisation is left for future work, but following the recommendations combined with sensible model inference is a potential way forward.
>
> > **Q2** “Do you have any thoughts about how your metric relates to other notions of fairness such as individual fairness metrics?”
>
> **A2** We briefly mention other fairness notions (L76-81). Individual fairness metrics aim to ensure similar treatment for similar individuals, but face challenges in defining similarity and computational complexity (Dwork, 2012; Kilbertus, 2018; Xiang, 2021). Our metric compares estimation error across legally protected attributes groups. It could be extended to consider estimation error across individuals or overlapping subgroups, engaging with other fairness metrics. Converting group-based metrics to individual assessments often reflects the same concerns (Binns, 2020).
>
> > **Q3** “Could you explain "taste-based discrimination" further?”
>
> **A3** Taste-based discrimination is an economic concept of discrimination where unequal treatment is based on the personal prejudices or preferences of decision-makers toward certain groups regardless of objective factors (Becker, 1957). In Section 2.5, in our decision-theoretic framework, it could arise if the utility function unjustifiably disfavors a group based on protected attributes. This concept highlights how discrimination can occur in automated decision-making systems, even when the predictive model itself is unbiased. By formalising decision-making in the decision-theoretic framework, our approach can distinguish between different sources of discrimination, including taste-based.
>
> ## Limitations
>
> We are happy to expand our limitations section. Could you clarify which issues should be more prominently discussed in Section 1.4?
>
> ## Summary
> We appreciate your constructive questions and feedback. With these clarifications, would you be willing to improve your score?

---

> > ### Comment · Reviewer_XoDZ · 2024-08-13
> >
> > Thank you for your detailed rebuttal and clarifications and apologies for the late response.
> >
> > I appreciate that this is theoretic work and not applied, however I would expect when talking about the law and fairness there should be some way for the reader to understand how it might be useful in practice. I think adding the existing case study throughout the text will help this. Given this addition and the clarifications on the contributions, I am happy to increase my score as I think this work could be valuable for the Neurips and fairness community.
> >
> > As for adding to the limitations section (Section 1.4) - I would make it clear about the difficulties of finding the true model, or in other words stress the inherent challenges in evaluating the estimation error.

---

### Official Review · Reviewer_iALf · 2024-07-12

**Soundness:** 4
**Presentation:** 3
**Contribution:** 3
**Rating:** 7
**Confidence:** 4

**Summary:**

This paper provides a UK-and-European-law-based view of anti-discrimination law as it relates to fair machine learning and automated decision systems. It does a good job laying out the doctrine, arguing correctly that work in this area to-date is very centered on US legal concepts such as disparate treatment vs. disparate impact. Although I am willing to believe that there are subtle differences that drive important aspects of fair ML analysis, as the paper claims, I think the specifics of these differences could be made much clearer and need to be for the paper to have the impact it should.

Of particular note, the paper is very well situated in the surrounding literature. Although this contextualization should make the contributions more clearly offset from prior work, as presented I find the opposite: it is difficult to tell what is new as a contribution here. For example, while the contributions are clearly identified in 1.4, I think it would aid the paper if they appeared higher in the intro and were clearer about what is new and why it matters. The example in Appendix A could be used as a running example to show where new concepts are needed and what about existing work does not capture this different legal regime. In particular, after claiming that disparate treatment/disparate impact are distinct to direct & indirect discrimination, the definitions given from 105-114 seem to align tightly to the former. And while I'm not a lawyer, I don't believe that disparate impact claims require a showing of intent under US law either, so I found that distinction somewhat confusing.

On the technical level, the discussion of the true data generating process should really be contextualized in the literature on measurement and construct validity, specifically with respect to work by Jacobs & Wallach, which in particular encompasses the material in 2.3 on estimation parity (at least in part). Also, the causal analysis components of the discussion of data generation could cite more of the work of Kohler-Hausmann and also Hu (one paper from these authors is cited, but others are also relevant and speak more directly to causality and counterfactual fairness claims).

As a final observation, although the ML community talks in terms of "fair" outcomes, it is often conceptually clearer (and more in line with legal analysis) to use the same techniques as tools for identifying "unfair" activities or outcomes. Phrasing some of the claims this way may condense some arguments and tighten the presentation overall. Related to this, the discussion of these tools as part of an overall practical strategy for risk management is important and should receive more attention. For example, it would be good to discuss how the measures proposed would be used in real legal analysis of an example, such as in litigation or a regulatory proceeding.

I was also a bit confused about the analysis of constructed proxies for protected variables in 2.7. I understand that it's necessary to look beyond a formalistic view of whether a specific attribute is considered, but what happens if the proxy for a protected attribute is (say) the sum of two legitimate attributes? Why is it good enough to use only legitimate features? Also, at 393-394 it might be valuable to look at the recent paper on "Less Discriminatory Algorithms" and compare the approaches and outlooks.

Incredibly minor:
* There is a missing period at 81.
* At 284-288, there is a latent call to questions of ecological validity which could be made more explicit

**Strengths:**

* Generalizing beyond the US legal context is important and valuable and this paper does a good job explaining the UK and related legal systems' approach to anti-discrimination law.
* The paper is well written and well situated in existing literature

**Weaknesses:**

* Novelty is at times hard to identify. I think it's there, but the claims on what it covers should be clearer. In particular, the discussion of the decision-theoretic framing seems a bit under-attended even though it's potentially very useful.
* Some important concepts are missed, notably theories of measurement and construct validity/reliability are at least partially re-invented when they should just be treated as background.

**Questions:**

* Can the example in Appendix A be a running example?
* What is new over and above existing literature on ML fairness? How can this novelty be more clearly offset in the presentation? Here I think specifically about the conclusions at 415-430, which seem rather anodyne in light of existing literature.

**Limitations:**

I believe the limitations are expressed well.

---

> ### Author Rebuttal · Authors · 2024-08-06
>
> Thank you for your positive feedback, detailed review, and suggestions for improving our paper. We appreciate your recognition that our paper **"does a good job laying out the doctrine"**,  **"is well written and well situated in existing literature"** and that **"generalizing beyond the US legal context is important and valuable."**
>
> ## General
>
> We will address all points raised in your summary, including:
>
> 1. Contributions and novelty: We will clarify our novel contributions earlier in the paper, more explicitly demonstrating their importance for fair ML in the context of UK laws (and European Union, many Commonwealth, and other common law jurisdictions).
> 2. Contextualise within existing literature: We will expand our discussion on measurement and construct validity, and causality in fairness. Jacobs and Wallach (2021) are important as they closely align with the idea of a latent data-generating process.
> 3. US vs UK definitions: We will clarify the confusing language at L104. At present, it leads directly to similar definitions under US and UK law but our comment on differences is not specific to the definitions. We will amend to explain the differences between evidentiary burdens, causality tests, justifications, defences, and other practical aspects discussed throughout the paper. Additionally, the comment on intention is with respect to direct discrimination, which differs from US disparate treatment that often requires intention.
> 4. Framing of fair vs unfair: We appreciate your suggestion and will reframe discussions in terms of “unfair” outcomes to clarify these arguments.
> 5. Proxy analysis: We will expand our discussion on constructed proxies. Proxy discrimination in the context of direct discrimination relates only to variables that are "indissociable" from the protected characteristic, so it is difficult to see how it could be the sum of two legitimate variables. If two variables together constitute a protected attribute for the purpose of indirect discrimination, the question arises whether the use of both variables causes the protected group to be put at a particular, unjustified disadvantage when compared to those without such attribute. Even if two legitimate factors act as a proxy for a protected attribute, it doesn't necessarily mean the system using these factors is illegitimate or discriminatory. The key considerations are included in our paper, including the tests for legitimacy, context, causality, and the UK effects-focused analysis.
> 6. Less discriminatory algorithms paper: We will incorporate the recent "Less Discriminatory Algorithms" paper, which provides a useful analysis of model multiplicity and choosing less discriminatory algorithms. The legal framework for identifying less discriminatory _alternatives_ is both less burdensome and broader in the UK than the US, such that it will likely align with our conclusion that in some instances the best approach might be to not use a model at all.
>
> ## Weaknesses
>
> > **W1** “Novelty is at times hard to identify” and the “discussion of the decision-theoretic framing seems a bit under-attended even though it's potentially very useful.”
>
> **R1** We would appreciate clarification on why you think the decision-theoretic framework is under-attended?
>
> Our main novelty and contribution lies in bridging the gap between UK anti-discrimination law and fair machine learning techniques. Specifically:
> 1. We provide the first comprehensive analysis of UK (and, in practice, EU, many Commonwealth and common law) anti-discrimination laws in the context of automated decision-making systems.
> 2. We introduce new fairness metrics (estimation parity and conditional estimation parity) that align with UK legal principles. Unlike much of the algorithmic fairness literature, which assumes the absence of estimation error or assumes the true causal structure is known, we analyse the legal role of the Data Generating Process (DGP) and introduce fairness metrics that account for estimation error. Importantly, these fairness metrics are grounded in anti-discrimination law, unlike many previous fairness metrics.
> 3. We move beyond the focus on group parity that dominates existing algorithmic fairness literature. Our paper explains that identifying disparity between groups is only one aspect of examining a case for unlawful discrimination. Even when disparity is important (in establishing evidence for a prima facie case), courts acknowledge that treating all groups the same can actually disadvantage a protected group and minimise important structural and true differences. Our formalisation allows for true differences to be justified, overcoming this limitation in existing approaches.
> 4. We offer a novel decision-theoretic approach to analysing discrimination. It includes a unique discussion of legal causation and the utility function, which has rarely been explored in previous work. It allows for a more nuanced understanding of how bias can arise in automated systems.
> 5. We provide the first discussion of legitimate aims in the context of UK anti-discrimination law as it applies to automated decision-making systems.
> 6. Our legally informed definitions of legitimate \(x\) variables are unique to UK anti-discrimination law and provide a new framework to assess justifiable features for ML.
>
> To make this novelty clearer, we will:
> - Highlight these contributions in the abstract and earlier in the introduction.
> - Use Appendix A as a running example to illustrate how our framework provides insights that traditional approaches might miss, particularly regarding estimation error and the justification of true differences.
> - Articulate how our decision-theoretic framework, analysis of the DGP, and consideration of legitimate aims provide a more comprehensive approach to addressing unlawful discrimination than existing methods.

---

> > ### Comment · Reviewer_iALf · 2024-08-14
> >
> > I don't have any other comments or questions, and very much appreciate the responses to the review. I remain very happy with this paper and believe the revision will address the smaller issues raised in all the reviews.
> >
> > I leave as a comment to the area chairs that the issues I raise and the issues raised by reviewer 4EE7 are very similar but our scores are very different. To this, I offer that while the recommendations are very general, using a case study through the paper to answer the critical question of why the difference in legal regimes matters in practice shows why my outlook is positive (also, I believe that contributions need not be a definition or a model, but could be a review of relevant law that unpacks the requirements on how existing work can be applied in a different and understudied context).
> >
> > On the point about making more use of the decision-theoretic framework, adding material about choices between available models and incorporating a running example provides the opportunity to show how the decision-maker's choices can be thought of as decision-theoretic optimization under the legal constraints described here. I believe that change is feasible straightforwardly, since the necessary material is all in the paper (just some is in the appendix).
> >
> > Hu gives commentary on earlier work of Kohler-Hausmann here: https://www.phenomenalworld.org/analysis/disparate-causes-i/. You are likely also aware of another paper focusing specifically on the construction of sensitive groups and the structuring of the data generating process in which they collaborated: https://dl.acm.org/doi/abs/10.1145/3351095.3375674. The general point is that getting at the question of which group differences are causally meaningful vs. which are protected by anti-discrimination law requires either importing an exogenous ontology of protected groups (which the law provides but may not define in as much detail as is needed) or ignoring the contextual construction of subgroup structure in a population for a use case. There are many deep philosophical questions here, and this paper can't reckon with all of them, but it's important to point out where the lines are still blurry to make the scope as clear as possible

---

> ### Author Response · Authors · 2024-08-06
> **Rebuttal by Authors (2/2)**
>
> > **W2** “Some important concepts are missed, notably theories of measurement and construct validity/reliability are at least partially re-invented when they should just be treated as background.”
>
> **R2** We will expand our discussion to include references to relevant work on measurement and construct validity, particularly based on Jacobs & Wallach, incorporating in Section 2.3. We will also incorporate more of Kohler-Hausmann and Hu’s research on causality. Could you please specify which papers by Kohler-Hausmann and Hu you believe are most relevant? We're familiar with the paper by Kohler-Hausmann and Dembroff on causality in the US context, which we could reference in a comparative sense.
>
> ## Questions
>
> > **Q1** “Can the example in Appendix A be a running example?
>
> **A1** Yes, we fully agree with this suggestion and will include specific references to the case study throughout the text. We believe this will also aid the understanding of the practical use of these formalisations and how the measures would be used in real legal analysis.
>
> > **Q2** “What is new over and above existing literature on ML fairness? How can this novelty be more clearly offset in the presentation?”
>
> **A2** Please see comment in **R1** above.
>
> ## Limitations
> Thank you for confirming that “the limitations are expressed well.”
>
> ## Summary
> We appreciate your thorough feedback and are committed to improving the paper based on your valuable suggestions. We hope you will continue to support our paper for acceptance. Do you have any further questions or comments?

---

### Decision · Program_Chairs · 2024-09-25

**Decision:**

Reject

**Comment:**

This paper explores the legal challenges related to algorithmic fairness from the perspective of UK anti-discrimination law. Most reviewers are positive about the work. However, concerns have been raised regarding the novelty and technical contributions of this study. Additional comments were provided during the discussion period, highlighting several key points:

* Much of the content appears to be a review of existing literature, a point already noted in two reviews. Some of the comments suggest that the considerations and recommendations provided in the paper are not novel.

* It is generally hard to identify practical benefits of this work. It is important for the paper to clearly explain how systems developed under American anti-discrimination laws could contravene UK legislation, specifically pointing out the differences between US and UK anti-discrimination laws that impact ML practitioners.

* The paper would benefit from positioning its contributions relative to existing definitions and concepts of fairness, clarifying why UK anti-discrimination law's requirements are not addressed by these concepts.

* Although the paper could be of interest to researchers working on algorithmic fairness, its accessibility to the broader NeurIPS audience is questionable. The case study in the appendix could improve clarity, though it is unclear if this would necessitate a major rewrite.

Many of these concerns appear to be reasonable, and it is generally hard to evaluate whether the authors would adequately address these concerns without seeing a revision. Hence, I strongly encourage the authors to try to revise their paper accordingly.